# Gαq-Stimulated Gene Expression Is Insensitive to Bromo Extra Terminal Domain Inhibitors in HEK 293 Cells

**DOI:** 10.3390/ijms26188904

**Published:** 2025-09-12

**Authors:** Ashika Jain, Viviane Pagé, Dominic Devost, Darlaine Pétrin, Terence E. Hébert, Jason C. Tanny

**Affiliations:** Department of Pharmacology and Therapeutics, McGill University, Montreal, QC H3G 1Y6, Canada; ashika.jain@mail.mcgill.ca (A.J.); viviane.page@mcgill.ca (V.P.); dominic.devost@mcgill.ca (D.D.); darlaine.petrin@mcgill.ca (D.P.)

**Keywords:** bromodomain and extraterminal domain (BET) family proteins, DREADDs, cellular signaling, G proteins, transcriptional regulation

## Abstract

Bromodomain and extraterminal domain (BET) family proteins are ubiquitous transcriptional co-activators that function broadly in cellular differentiation, proliferation, and stress responses. Pharmacological inhibition of BET proteins with small molecules that disrupt bromodomain engagement with acetyllysine residues (such as JQ1) or drive their degradation through the ubiquitin–proteasome system (such as dBET6) ameliorates pathological gene expression in a range of systems and shows promise as a potential therapeutic strategy. Understanding the cell-type and signaling pathway requirements that dictate BET dependence in a particular cellular context remains incomplete. We previously demonstrated that, in neonatal rat cardiomyocytes, GPCR-induced hypertrophy responses depended strongly on the BET protein Brd4 when signaling was coupled to Gαs, but not Gαq. Here, we tested whether Brd4 was differentially responsive to G protein isoforms in HEK 293 cells by expressing Gαs- or Gαq-coupled Designer Receptors Exclusively Activated by Designer Drugs (DREADDs). Gαq induced the expression of a group of early response genes and inflammatory genes in a manner largely insensitive to pharmacological BET inhibition, consistent with our previous data in cardiomyocytes. Gαs activated a small subset of the Gαq-induced genes, but this effect was largely reversed by dBET6. Our data further suggest that there may be general signaling requirements to activate Brd4 across cell types.

## 1. Introduction

The bromodomain and extraterminal domain (BET) families of transcriptional co-activators are integral components of both physiological and pathological gene expression programs. BET proteins have been studied extensively as drivers of cancer cell proliferation, both for hematological malignancies and solid tumors [1,2]. BET proteins are also required for inflammatory gene expression, pathological cardiac hypertrophy, and spermatogenesis [3,4,5,6,7,8]. The BET family consists of Brd2, Brd3, and Brd4, all of which are broadly expressed in mammalian tissues, as well as the testis-specific variant BrdT. Brd4 is thought to be the relevant player in most of the functions mentioned above [1,2]. All BET proteins contain two tandem bromodomains, which interact with acetyllysines on histones and other regulatory proteins [1]. The extraterminal domain binds to multiple chromatin-modifying enzymes, although the regulatory significance of these interactions is not understood [9,10]. A C-terminal domain in Brd4 and Brdt binds to the positive transcription elongation factor b complex P-TEFb [11]. Brd4 also associates with the Mediator complex and participates in the recruitment of RNA polymerase II (RNAPII) to promoters and enhancers throughout the genome [10,12,13]. Thus, BET proteins, and Brd4 in particular, act broadly to affect multiple steps in RNAPII transcription.

Interest in BET proteins has been spurred by the development of small molecules that bind to the BET bromodomains, thereby competing with acetyllysine and diminishing BET chromatin occupancy. These BET inhibitors (such as JQ1, iBET-151, or RVX-208) have shown beneficial effects in cell culture and animal models of multiple types of cancer, inflammatory disease, and cardiac hypertrophy [14,15,16,17,18]. More recently, they have been converted into effective BET degraders (such as dBET6) through chemical linkage to small molecules that bind to the VHL or cereblon E3 ubiquitin ligases [19,20]. It is critical to further define how BET proteins act in transcription to guide the future use of these potential therapeutics.

Despite their connections to multiple generally acting components of the RNAPII transcriptional machinery, BET proteins have gene-selective roles in transcription that are cell-type specific and remain poorly understood. Studies in cancer cell lines have demonstrated that the sensitivity of particular cell lines to the growth-restricting effects of BET inhibitors correlated with the activity of super enhancers proximal to key oncogenic driver genes (such as c-myc) [1,21,22]. As originally defined, super enhancers are distinguished from conventional enhancers by virtue of their large size (often 20 kb in length or more), as well as their association with exceptionally high levels of Mediator and Brd4 [22,23,24]. However, it is unclear whether similarly designated super enhancers are important for other transcriptional responses that require Brd4 [25,26,27]. Brd4 action at specific promoters or enhancers may instead be dictated by interactions with DNA-binding transcription factors [28,29,30]. BETs are also regulated by post-translational modification. Brd4 is hyperphosphorylated in several cancer cell lines, and phosphorylation by multiple cyclin-dependent kinases or by casein kinase II has been associated with enhanced chromatin binding [28,31,32]. Such a mechanism is also implicated in the Brd4-dependent activation of immediate early genes in cortical neurons [33]. Cytokine signaling leads to JAK2-dependent phosphorylation of Brd4 in colorectal cancer cells, thereby preventing Brd4 protein degradation by the ubiquitin proteasome system and enhancing the transcription of key oncogenic genes [34]. BET protein stability is regulated by polyubiquitylation in multiple cell types; the altered regulation of these pathways in cancers has been associated with resistance to BET inhibitors [35,36,37]. Thus, BET proteins are dynamic regulators that are highly responsive to inputs from cellular signaling pathways.

We previously demonstrated that in a cellular model of neonatal rat cardiomyocyte hypertrophy, Brd4 activity was differentially sensitive to two hypertrophic stimuli acting through different G protein-coupled receptors (GPCRs) [38]. If hypertrophy was induced by activation of the endothelin A (ET_A_) receptor using the ligand endothelin-1, it proceeded relatively unimpeded by the BET inhibitor JQ1. However, stimulation of hypertrophy through the activation of the α1-adrenergic receptor with the agonist phenylephrine was JQ1-sensitive. Concordantly, Brd4 occupancy at promoters and enhancers of hypertrophy-induced genes was enhanced by phenylephrine, but not by endothelin-1. These differences appeared to be due to the distinct signaling pathways activated by the two receptors: the ET_A_ receptor is coupled to the Gαq G protein isoform, whereas the α1-adrenergic receptor is coupled to Gαq and Gαs [38,39]. Gαs-dependent signaling appeared to be important for Brd4 activation, as Brd4 occupancy in chromatin immunoprecipitation (ChIP) experiments could be enhanced by stimulating adenylate cyclase or blocked by the inhibition of protein kinase A (PKA) [38].

Based on these results, we hypothesized that Brd4 participation in GPCR-driven transcriptional responses would necessitate signaling through Gαs, and that responses following Gαq signaling would be relatively Brd4-independent. We sought to test this using a tractable model cell line and engineered GPCRs that couple to defined G protein subtypes. The results were consistent with our hypothesis and may help to define cellular contexts that dictate responsiveness to BET inhibitors.

## 2. Results

### 2.1. Functional Validation of DREADD Constructs

We used a Gαq-DREADD and a Gαs-DREADD to confer selective G protein coupling [40,41,42]. We expressed each DREADD in HEK 293 cells to interrogate the effects of these signaling pathways on gene expression. To verify that the DREADDs were functional in our system, we tested doses of deschloroclozapine (DCZ), a potent DREADD ligand, over a 5-log range in cells expressing one or the other DREADD as well as bioluminescence resonance energy transfer (BRET)-based biosensors that monitor either Gαq or Gαs-coupled signaling [39,43]. BRET was detected in cells expressing Gαq-DREADD and a PKC biosensor, but not in cells expressing the biosensor with an empty vector (Figure 1A). Similarly, BRET was detected in cells expressing Gαs-DREADD and the EPAC biosensor, but not in cells expressing the biosensor with an empty vector (Figure 1B). The response to DCZ for either DREADD was detected over control at 100 nM at 10-, 20-, or 30 min treatment times (Figure 1A,B). Responses appeared to peak at 1 mM for the 20 and 30 min treatments; the 1 mM responses were sustained for up to 1 h for both DREADDs (Figure 1C).

### 2.2. Confirmation of BET Inhibitor Activity

Next, we verified the effectiveness of standard BET inhibitors in our system. Treatment of DREADD-transfected HEK 293 cells with 100 nM dBET6 (a dose based on previous studies; [20]) led to ~75% depletion of Brd4 protein after 3 h, as assessed by immunoblot (Figure 2A). Immunoblotting confirmed that dBET6 was similarly effective in the presence or absence of DCZ in DREADD-expressing cells (Appendix A). We used c-myc mRNA levels to assess the efficacy of 1 mM JQ1 treatment, as c-myc transcript levels are sensitive to JQ1 in several transformed cell lines [21,44]. Unexpectedly, c-myc transcripts increased over time upon JQ1 treatment (Figure 2B). We attribute this effect to P-TEFb activation driven by JQ1-dependent release of Brd4 from chromatin, a mechanism proposed to account for JQ1-dependent reversal of HIV latency and JQ1-dependent immediate early gene activation in striatal neurons [45,46]. That dBET6 treatment, which depletes Brd4 protein, did not lead to any change in c-myc transcripts, is consistent with this interpretation (Figure 2B). Although we did not investigate the mechanism for this effect further, our results are consistent with the notion that JQ1 inhibited BET association with chromatin.

### 2.3. Analysis of G Protein Activation by RNA Sequencing

We then performed RNA-seq in DREADD-expressing cells in the presence or absence of DCZ, and/or BET inhibitor. We chose 1 h DCZ treatments to capture cells when G protein signaling was still ongoing (see Figure 1) and when differentially expressed genes are likely to be detected. These treatment conditions were carried out either alone or in combination with 3 h inhibitor treatments (such that DCZ was added in the last hour of inhibitor treatment). A control sample that was not treated with DCZ or an inhibitor was also included for each DREADD. We analyzed two biological replicates of each condition using RNA-seq. We observed high concordance between replicates, as indicated by Pearson correlations and principal component analysis (Appendix A).

DCZ treatment significantly increased the expression (by 1.5-fold or more; *p* adj ≤ 0.1) of 55 genes in cells expressing Gαq-DREADD (relative to the DMSO control) and 4 genes in cells expressing Gαs-DREADD. Interestingly, no transcripts were significantly decreased in either condition (Figure 3A). The 55 genes induced by Gαq signaling included a set of 22 genes classified as primary response genes upon ERK activation (Figure 3B) [47]. This was expected, given that Gαq signaling is known to increase ERK1/2 activation in HEK 293 cells [48]. In addition, KEGG pathway mapping showed significant enrichment of functions related to IL-17, TNF, and NFκB signaling pathways among Gαq-induced genes (Figure 3C).

The four genes (PCK1, DUSP1, FOS, NR4A2) that were significantly upregulated by DCZ in cells expressing Gαs-DREADD were also induced by Gαq-DREADD, and two of them (DUSP1, FOS) were primary response genes (Figure 3A) [47]. This suggests a shared function for Gαq and Gαs signaling in activating ERK1/2 in HEK 293 cells.

### 2.4. Effects of BET Inhibition on Gαq and Gαs-Mediated Gene Expression

Next, we tested the impact of BET inhibitors on DREADD-induced gene expression changes. Treatment with dBET6 alone (i.e., no DCZ) decreased levels of 2140 transcripts and increased levels of 552 transcripts in cells expressing Gαq-DREADD (Figure 4A). Very similar changes were observed in cells expressing Gαs-DREADD, although the genes upregulated in dBET6-treated cells expressing either DREADD were only ~50% overlapping (Appendix A). That the DREADDs alone showed slight differences in the presence of dBET6 may indicate that they have subtle effects on signaling in the absence of ligand, as has previously been shown for Gαs-DREADD in some cell types [41]. Importantly, neither DREADD alone had a significant impact on gene expression in HEK 293 cells, as no differentially expressed genes were identified when comparing the RNA-seq datasets for the two DREADDs in the absence of DCZ.

The largely inhibitory effect of dBET6 on transcript levels is consistent with the established function of Brd4 and other BET proteins in transcriptional activation [1]. KEGG pathway analysis of the downregulated genes showed significant enrichment of gene groups linked to signaling and transcriptional regulation in cancer; no significant associations were found among the upregulated genes (Figure 4B). JQ1 treatment of Gαq-DREADD-expressing cells also altered transcript levels for hundreds of genes, with a more even distribution between upregulation and downregulation (1324 and 859, respectively; Figure 4C). The changes were very similar in cells expressing Gαs-DREADD, with downregulated genes again showing a more complete overlap than upregulated genes (Appendix A). The transcripts decreased by JQ1 were a significantly overlapping subset of those decreased by dBET6 (Figure 4D; shown is data for cells expressing Gαq-DREADD), and KEGG pathway analysis showed significant enrichment of the same functional classes of genes downregulated by dBET6 (Figure 4E). There was less pronounced overlap between the transcripts upregulated by JQ1 and dBET6 (Figure 4D). Interestingly, KEGG pathway analysis of the JQ1-upregulated genes revealed significant enrichment of genes related to “transcriptional misregulation in cancer,” consistent with our observation that c-myc transcript levels were enhanced. HEXIM1 was also significantly induced by JQ1 treatment (Figure 4C). HEXIM1 encodes a component of the inactive P-TEFb complex induced upon exposure to stresses that mobilize P-TEFb from its inactive to its active state, thus serving as a feedback regulator of these responses. This is consistent with a mechanism whereby JQ1-dependent release of Brd4 from chromatin mobilizes a fraction of the inactive P-TEFb pool to enhance the transcription of some genes [45]. The group of transcripts whose levels were increased by JQ1, but not by dBET6, is likely to be regulated by a similar mechanism.

When compared to vehicle-treated DREADD-expressing cells, cells treated with both a BET inhibitor and DCZ showed upregulation and downregulation of sets of genes that overlapped significantly with the same sets identified in cells treated with the BET inhibitor alone (Appendix A). This is consistent with the large gene regulatory effect of BET inhibition compared to that of DREADD activation in this cell line. However, we observed that the combination of DCZ and BET inhibitors consistently led to larger numbers of affected transcripts (up- or downregulated) than BET inhibitors alone (Appendix A). This was most evident for transcripts showing increased expression in Gαs-DREADD-expressing cells: 638 genes were identified upon treatment with DCZ and dBET6, whereas 419 were identified with dBET6 alone (Appendix A). Thus, although DREADD activation by itself affected the expression of small numbers of genes, it modulated the larger effects of BET inhibition.

We found that DCZ treatment of Gαq-DREADD-expressing cells in the presence of either dBET6 or JQ1 induced the expression of most of the genes induced in the absence of BET inhibitors. In the presence of dBET6, DCZ induced the expression of 49 genes, 38 of which were also induced by DCZ in the absence of the BET inhibitor (Figure 5A,B). DCZ treatment did not downregulate any transcripts in this context, as we observed in the absence of BET inhibitors. A total of 39 genes were induced by DCZ in the presence of JQ1, 36 of which responded to DCZ alone (Figure 5C,D). Thus, 65–70% of the genes activated by Gαq signaling in HEK 293 cells are insensitive to BET inhibition. KEGG pathway analysis of the groups of genes resistant to JQ1 or dBET6 showed that they were significantly enriched for functional categories related to cytokine signaling, similar to what we observed in the absence of the BET inhibitor (Appendix A and Figure 3A). This indicated that the character of the Gαq gene expression response was not altered by BET inhibitors.

Three of the four genes induced by DCZ treatment of cells expressing Gαs-DREADD (PCK1, NR4A2, and FOS) were not induced in the presence of dBET6 (Appendix A). Interestingly, these three genes were among those that were resistant to dBET6 in Gαq-DREADD-expressing cells (Figure 4A). All four genes were induced by DCZ in the presence of JQ1, as well as eight additional genes (Appendix A). The divergent effects of dBET6 and JQ1 in this context may again reflect the altered function of BET proteins released from chromatin by JQ1 (see Figure 2B). Although based on a small number of genes, these results suggest that Gαq and Gαs signaling led to the induction of overlapping sets of genes through distinct mechanisms.

## 3. Discussion

In this study, we expressed DREADDs in HEK 293 cells to specifically assess Gαq- or Gαs-dependent changes in gene expression and assess how BET inhibitors influenced such changes. Although the scope of the gene expression changes associated with DREADD activation in our system was limited, our data point to transcriptional induction by Gαq signaling as largely independent of BET co-regulators. This is consistent with the notion that BET proteins are context-specific regulators that respond to cellular signaling inputs, rather than acting as general transcription factors.

DREADDs have been used to study cellular signaling events extensively in a variety of cell types [42]. To our knowledge, this is the first example of the use of DREADDs to specifically interrogate G protein-mediated changes in gene expression. Stimulation of Gαq-DREADD in HEK 293 cells activated a subset of immediate early genes. This was consistent with canonical Gαq signaling involving phospholipase Cβ, Ca^2+^ mobilization, protein kinase C (PKC), and subsequent ERK1/2 activation [49,50]. We also observed an increased expression of a subset of genes that are normally targets of inflammatory signaling pathways, such as IL-17 and TNFα. This subset included a group of linked CXCL chemokine genes co-regulated by a proposed super enhancer [51]. These effects could reflect Gαq-dependent activation of NF-κB through phosphoinositol-3-kinase (PI3K) and Akt-mediated phosphorylation of IκB kinase [52]. Deciphering the nature of pathways preferentially activated by stimulation of Gαq-DREADD in HEK 293 cells will require further analysis.

Stimulation of Gαs-DREADD induced the expression of only four genes, all of which were among those induced by Gαq-DREADD. Gαs signaling is also associated with ERK1/2 activation in HEK 293 cells, and so an effect on immediate early genes that overlaps with that of Gαq signaling might be expected [48,49,53]. It is unclear why so few responsive genes were identified upon stimulation of Gαs-DREADD. We are currently investigating whether this could reflect a rapid burst of gene expression change induced by Gαs signaling that we did not capture in our RNA-seq analysis.

Consistent with previous studies, dBET6 treatment led to more robust effects on steady-state transcripts than did JQ1, and its effects were more biased toward reduced transcript levels. This is most likely due to the bromodomain-independent functions of BETs, which have been characterized for Brd4 and are mitigated by dBET6 but not by JQ1 [20,54]. We also observed a group of genes whose transcription was apparently stimulated by the release of BETs from chromatin, including c-myc and HEXIM1. This occurs because Brd4, which is liberated from chromatin by JQ1, is able to engage P-TEFb and stimulate transcription [45]. The effect of JQ1 on immediate early genes that we found here resembles our previous findings in striatal neurons but differs from that reported in cortical neurons [33,46]. Such differences may result from the differing status of cellular signaling pathways between cell types, consistent with the idea that BETs are signal transducers.

In cells expressing Gαq-DREADD, DCZ activated gene expression largely independently of BETs. This was evident from the fact that 65–70% of the DCZ-responsive genes were still induced in the presence of either JQ1 or dBET6. Perhaps more striking was the fact that Gαq signaling activated genes linked to cytokine signaling and NFκB even in the presence of BET inhibitors. Inflammatory genes are highly sensitive to BET inhibition in several physiological contexts; the cluster of chemokine genes activated by Gαq signaling in our dataset was previously found to be JQ1-sensitive in liver endothelial cells [3,7,51]. Our previous findings in primary cardiomyocytes indicated that inflammatory genes were insensitive to JQ1 when cells were stimulated with endothelin-1, a ligand that signals through a Gαq-coupled GPCR [38]. The findings here again suggest that Gαq-coupled GPCR signaling elicits transcriptional responses that do not strictly require BET protein function. This may reflect a general property of Gαq-coupled signaling pathways, given that we find evidence for it in two disparate cell types (HEK 293 cells and primary cardiomyocytes). Employing DREADDs to activate specific Gα pathways in additional cell types beyond the HEK 293 model will be needed to establish this principle. The fact that only four genes were induced by DCZ in the presence of Gαs-DREADD precluded a robust comparison between the gene expression effects of the two signaling pathways. However, all four genes were induced by Gαq signaling as well. In this case, the genes were BET-inhibitor-resistant, whereas DCZ-mediated induction of three of the four genes was blocked by dBET6 in the presence of Gαs-DREADD. This is again reminiscent of our previous findings in cardiomyocytes: when a Gαs-coupled GPCR was stimulated, inflammatory genes were induced in a JQ1-sensitive manner.

BET inhibitor resistance linked to constitutively active JAK2 or mutations in the cullin E3 ligase SPOP is caused by stabilization of Brd4 protein, leading to increased levels that counteract BET inhibition [34,35,37]. We did not observe significant effects of DREADD signaling on Brd4 protein levels in the presence of dBET6, suggesting that Brd4 stabilization does not account for BET inhibitor resistance in our system. Resistance may result from alleviating endogenous BET inhibitory mechanisms, such as binding by the retinoblastoma protein RB [55]. Alternatively, Gαq-DREADD may trigger a BET-independent pathway for inflammatory gene activation, analogous to Wnt pathway activation in BET-inhibitor-resistant leukemias [56]. This latter mechanism would be consistent with our previous findings in cardiomyocytes, which showed no change in Brd4 levels upon Gαq-coupled or Gαs-coupled receptor stimulation but differences in Brd4 recruitment to target loci by chromatin immunoprecipitation (ChIP) [38]. Future work will be needed to distinguish between these mechanisms in our system. This will be important in furthering our understanding of the response of BETs to cell signaling pathways and how this can be harnessed in the use of BET inhibitors as therapeutics.

## 4. Materials and Methods

### 4.1. Cell Culture and Drugs

HEK 293(PL) cells [57] were grown in Dulbecco’s Modified Eagle’s medium (DMEM) high glucose + 5% (*v*/*v*) fetal bovine serum + 1% (*v*/*v*) penicillin/streptomycin at 37 °C. JQ1 (Abcam, Cambridge, UK; ab141498) and dBET6 (MilliporeSigma, Oakville, ON, Canada; SML2683) were dissolved in dimethyl sulfoxide (DMSO) at 5 mM and 500 μM, respectively. Deschloroclozapine (DCZ; Hellobio, Princeton, NJ, USA; HB9126) was dissolved in sterile ddH_2_O at 10 mM.

### 4.2. Plasmids

Plasmids for the expression of Gαq-DREADD (pcDNA5/FRT-HA-hM3D) and Gαs-DREADD (pcDNA5/FRT-HA-rM3D) were obtained from Addgene (Watertown, MA, USA) [40,41]. Plasmids for the expression of the PKC and EPAC biosensors have been described previously [39]. Transfection into HEK 293 cells was performed with Lipofectamine 2000 (ThermoFisher; Waltham, MA, USA) as per the manufacturer’s instructions.

### 4.3. Bioluminescence Resonance Transfer (BRET) Biosensor Assays

Cells were transfected with Gαq-DREADD, Gαs-DREADD, or a vector control (pcDNA 3.1; [39]) in combination with either a PKC biosensor plasmid or an EPAC biosensor plasmid. After transfection, cells were plated at a density of 30,000 cells/well in a poly-L-ornithine-coated 96-well white bottom plate (ThermoFisher; Waltham, MA, USA, 236105) and incubated for 24 h. Media was removed from each well, cells were washed with 150 μL Krebs buffer, and 80 μL Krebs buffer was added to each well. Fluorescent substrate coelenterazine 400A (Cedarlane, Burlington, ON, Canada; 10 μL) was added to each well. BRET measurements were performed in a TriStar 2 Multimode Plate Reader (Berthold Technologies, Germany) as described previously [57]. Briefly, the plate was maintained at 28 °C for one hour, the last 5 min of which were used to take a basal reading in the 410–515 nm spectral range. DCZ or vehicle was then added, and further readings taken at 28 °C after the indicated time intervals.

### 4.4. BRET Analysis

BRET ratios were determined by calculating the ratio of the light emitted by GFP10 over the light emitted by the RlucII, and results were expressed as ΔBRET = BRETagonist − BRETbasal [58]. ΔBRET was calculated as follows:ΔBRET_Gαs_ = [(Gαs-DREADD-EPAC/DCZ) − (Gαs-DREADD-EPAC)] − [(pcDNA-EPAC/DCZ) − (pcDNA-EPAC)].ΔBRET_Gαq_ = [(Gαq-DREADD-PKC/DCZ) − (Gαq-DREADD-PKC)] − [(pcDNA-PKC/DCZ) − (pcDNA-PKC)].

An average of three technical replicates was used for each measurement. Data were plotted as ΔBRET versus log [DCZ] with GraphPad Prism 10.0.

### 4.5. Protein Extraction and Immunoblotting

Cells were seeded at 350,000 per well in 6-well plates. Following a 24 h incubation, cells were treated with 100 nM dBET6 or vehicle, washed twice with cold phosphate-buffered saline (PBS), and suspended in 200–400 μL cold RIPA buffer [1% NP-40, 50 mM Tris-HCl pH 7.4, 150 mM NaCl, 1 mM EDTA, 1 mM EGTA, 0.1% SDS, 0.5% sodium deoxycholate, 1 mM PMSF, protease inhibitor tablets (MilliporeSigma, Oakville, ON, Canada)]. Suspensions were transferred to 1.5 mL tubes, mixed, and incubated for 15 min on ice. Lysates were centrifuged at 14,000× *g* for 15 min at 4 °C. Protein concentrations of the supernatants were measured using a BCA kit (ThermoFisher; Waltham, MA, USA) as per the manufacturer’s instructions. Thirty μg of supernatant was subjected to SDS-PAGE on an 8% gel, and the gel was transferred to a nitrocellulose membrane as described previously [38]. Immunoblotting was performed with anti-BRD4 (ThermoFisher; Waltham, MA, USA, PA5856620), anti-β-tubulin (ThermoFisher; Waltham, MA, USA, 32-2600), or anti-GAPDH (ThermoFisher; Waltham, MA, USA, AM-4300) antibodies, as described previously [38]. Blots were imaged with a GE Amersham Imager 600 (Cytiva Life Sciences, Marlborough, MA, USA). Images were processed with ImageJ software (2.17.0).

### 4.6. RNA Analysis by RT-qPCR

Cells were seeded at 350,000 per well in 6-well plates. Following 24 h incubation, cells were treated with 1 μM JQ1 or vehicle and washed twice with cold PBS. Cells were lysed with TRI reagent^®^ (MilliporeSigma, Oakville, ON, Canada), and lysates were transferred to 1.5 mL tubes followed by extraction with bromochloropropane. After a 15 min incubation at room temperature, lysates were centrifuged at 13,000× *g* for 15 min at 4 °C. The aqueous phase was precipitated with an equal volume of isopropanol, and RNA pellets were collected by centrifugation at 13,000× *g* for 8 min at 4 °C. RNA pellets were washed with 70% ethanol and dissolved in RNAse-free ddH_2_O. The concentration of the isolated RNA was measured using a Nanodrop 2000 (ThermoFisher; Waltham, MA, USA) spectrophotometer. One μg of total RNA was used as template for cDNA synthesis with random hexamers and M-MLV reverse transcriptase as previously described [38]. For qPCR, cDNA was diluted 1:10 and amplified with primer pairs complementary to c-myc or GAPDH (sequences available upon request) using BrightGreen 2X qPCR Mastermix (Applied Biological Materials, Richmond, BC, Canada) and a ViiA 7 Real-Time PCR System (ThermoFisher; Waltham, MA, USA). The qPCR amplification curves were analyzed via the 2^−ΔΔCt^ method as described previously [38].

### 4.7. RNA-Sequencing

Cells were transfected with DREADD plasmids and seeded at 350,000 cells per well in 6-well plates. Following a 24 h incubation, cells were treated with 100 nM dBET6, 1 μM JQ1, or vehicle for 3 h. During the last hour of inhibitor treatment, DCZ (1 μM) or vehicle was added. Cells were washed twice with cold PBS, and total RNA was extracted using the Qiashredder Homogenization kit and the RNeasy Mini kit (Qiagen, Germantown, MD, USA) as per the manufacturer’s instructions. RNA concentrations were measured with a Nanodrop 2000 spectrophotometer, and RNA quality was assessed on an Agilent 2100 Bioanalyzer (Agilent Technologies, Santa Clara, CA, USA). RNA samples (100 ng) with a RIN score >7 were submitted to Genome Quebec for paired-end library preparation and sequencing. Libraries were generated from 100 ng of total RNA, and mRNA enrichment was performed using the NEBNext Poly(A) Magnetic Isolation Module (New England BioLabs, Ipswich, MA, USA). cDNA synthesis was performed with NEBNext UltraExpress RNA Library Prep Kit (New England BioLabs, Ipswich, MA, USA) as per the manufacturer’s recommendations. Libraries were quantified using the KAPA Library Quanitification Kits—Complete kit (Universal) (Kapa Biosystems, Wilmington, MA, USA). Average fragment size was determined using a Fragment Analyzer 5300 (Agilent Technologies, Santa Clara, CA, USA) instrument. The libraries were normalized and pooled and then denatured in 0.02N NaOH and neutralized using a pre-load buffer. The pool was loaded at 140 pM on an Illumina NovaSeq X Plus 25B lane (Illumina, San Diego, CA, USA) following the manufacturer’s recommendations. The run was performed for 2 × 100 cycles (paired-end mode) (150 base pair reads at a depth of ~25 million reads per sample for 24 samples). A phiX library was used as a control and mixed with libraries at 1% level. Program BCL Convert 4.2.4 was then used to demultiplex samples and generate FASTQ files.

### 4.8. RNA-Seq Data Analysis

FASTQ files were subjected to adaptor trimming, and FASTP (v0.23.4) was used to filter the low-quality and duplicate reads [59]. The sequences were then aligned to the *Homo sapiens* genome (GRCh38, NCBI # GCF_000001405.26) using STAR (v2.7.11b) [60]. Gene-level read count matrices were obtained using FeatureCounts (v2.0.1) [61]. Differential expression analysis was performed with DESeq2 (v1.42.1) [62]. Pearson correlations and principal component analysis plots to compare biological replicates were generated using DESeq2. Venn diagrams comparing differentially expressed genes were generated using a web-based tool (https://bioinformatics.psb.ugent.be/webtools/Venn/; accessed on 1 June 2024). Volcano plots representing differentially expressed genes were generated using ggplot2. KEGG pathway enrichment was plotted using GOplot (v1.0.2).

## Figures and Tables

**Figure 1 ijms-26-08904-f001:**
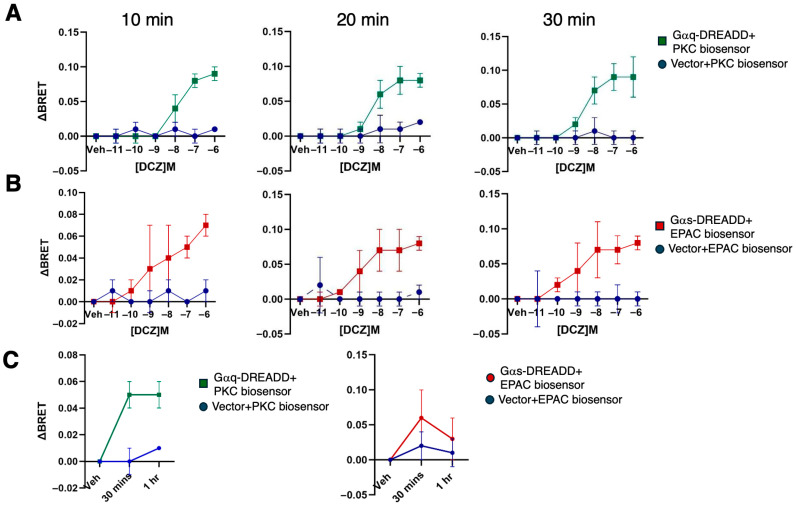
Functional validation of Gαq- and Gαs-coupled DREADDs using BRET biosensors. (**A**) DBRET was measured in HEK 293 cells transfected with the PKC biosensor and either G*α*q-DREADD or a vector control. Measurements were taken at the indicated doses of DCZ at the indicated times (n = 3 independent experiments; error bars indicate SEM). “Veh” indicates vehicle control for DCZ (ddH_2_O). (**B**) As in (**A**) for the EPAC biosensor and either Gαs-DREADD or vector control. (**C**) DBRET was measured in HEK 293 cells transfected with either DREADD/biosensor combination after 1 μM DCZ treatment for the indicated times (n = 3 independent experiments; error bars indicate SEM). “Veh” indicates 1 h vehicle control for DCZ (ddH_2_O) treatment.

**Figure 2 ijms-26-08904-f002:**
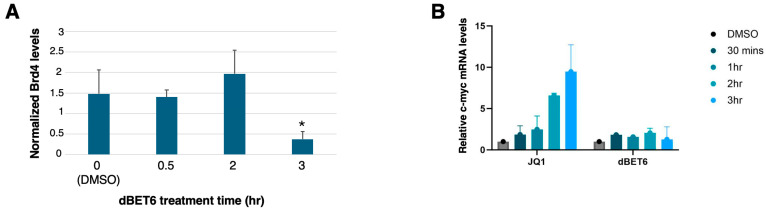
Functional validation of BET inhibitors. (**A**) Plot of ImageJ quantification of anti-Brd4 immunoblots (normalized to GAPDH loading control, n = 3; means ± SEM). The 3 h vehicle (DMSO) treatment is used as the “0” control. * indicates significant difference from 0 (two-tailed *t*-test; *p* < 0.05). (**B**) RT-qPCR to quantify relative c-myc transcript levels after treatment of HEK 293 cells with either 1 μM JQ1 or 100 nM dBET6 for the indicated times (n = 3 independent experiments). Values for 3 h vehicle (DMSO) treatment were set to 1.

**Figure 3 ijms-26-08904-f003:**
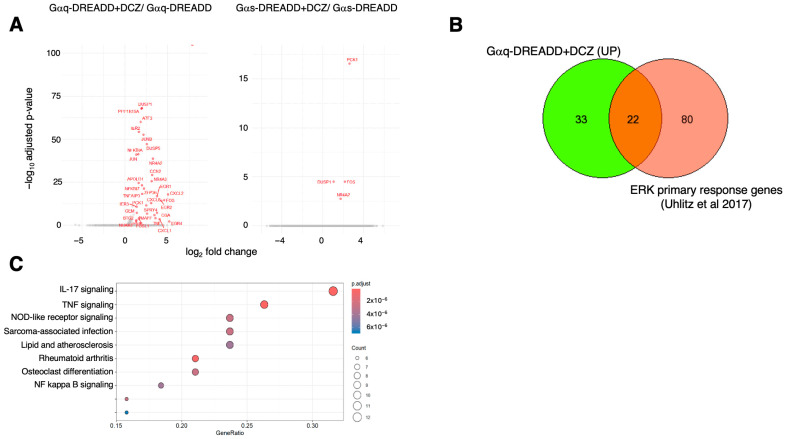
Gene expression changes triggered by DREADDs in HEK 293 cells. (**A**) Volcano plots of differentially expressed genes identified comparing cells expressing Gαq-DREADD (**right**) or Gαs-DREADD (**left**) ± DCZ treatment (fold change ≥ 1.5, *p* ≤ 0.1). (**B**) Venn diagram showing overlap of genes induced by Gαq-DREADD/DCZ and ERK primary response genes identified in [47]. (**C**) KEGG pathway analysis of 55 Gαq-DREADD/DCZ upregulated genes. “GeneRatio” indicates ratio of number of genes enriched in the indicated category to total number of differentially expressed genes.

**Figure 4 ijms-26-08904-f004:**
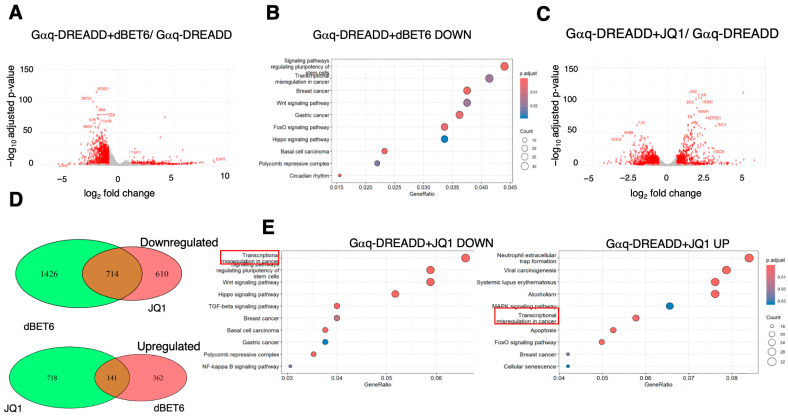
Gene expression changes triggered by BET inhibitors in HEK 293 cells. (**A**) Volcano plot of differentially expressed genes identified comparing cells expressing Gαq-DREADD ± dBET6 (fold change ≥ 1.5, *p* ≤ 0.1). (**B**) KEGG pathway analysis of significantly downregulated genes from (**A**). (**C**) Volcano plot of differentially expressed genes identified comparing cells expressing Gαq-DREADD ± JQ1 (fold change ≥ 1.5, *p* ≤ 0.1). (**D**) Venn diagrams comparing the significantly downregulated (**top**) and upregulated (**bottom**) genes identified upon dBET6 or JQ1 treatment. (**E**) KEGG pathway analysis of significantly upregulated (**left**) and downregulated (**right**) genes from (**C**).

**Figure 5 ijms-26-08904-f005:**
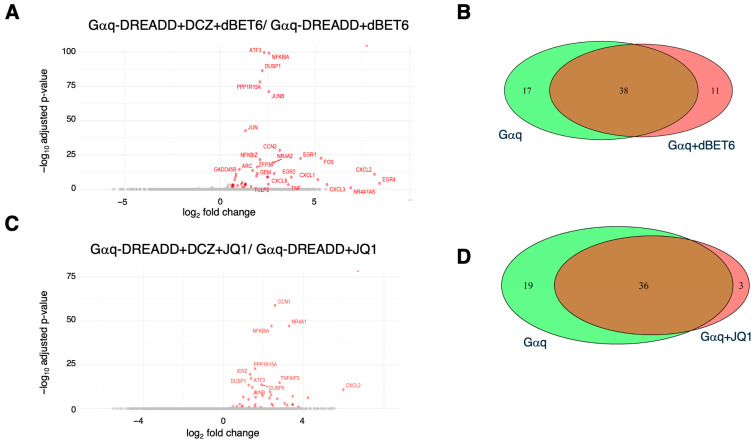
Most gene regulatory effects of Gαq-DREADD activation are maintained in the presence of BET inhibitors. (**A**) Volcano plot of differentially expressed genes identified comparing cells expressing Gαq-DREADD with combined DCZ/dBET6 treatment to those expressing Gαq-DREADD and treated with dBET6 alone (fold change ≥ 1.5, *p* ≤ 0.1). (**B**) Venn diagrams comparing the significantly upregulated genes identified upon DCZ treatment of cells expressing Gαq-DREADD in the presence or absence of dBET6. (**C**) Volcano plot of differentially expressed genes identified comparing cells expressing Gαq-DREADD with combined DCZ/JQ1 treatment to those expressing Gαq-DREADD and treated with JQ1 alone (fold change ≥ 1.5, *p* ≤ 0.1). (**D**) Venn diagrams comparing the significantly upregulated genes identified upon DCZ treatment of cells expressing Gαq-DREADD in the presence or absence of JQ1.

## Data Availability

RNA-seq datasets were deposited in the NCBI Gene Expression Omnibus (GEO) under accession GSE299127.

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
