# Peer review of "Gαq-Stimulated Gene Expression Is Insensitive to Bromo Extra Terminal Domain Inhibitors in HEK 293 Cells"

_ijms, 2025, doi:10.3390/ijms26188904_

Round 1

Reviewer 1 Report

Comments and Suggestions for Authors

I have read the manuscript with great interest. This manuscript by Jain et al. investigates the differential requirement of BET proteins in gene expression programs driven by specific G protein subtypes, Gαq and Gαs. The authors build upon their previous work in cardiomyocytes by using a more defined system: HEK 293 cells expressing Designer Receptors Exclusively Activated by Designer Drugs (DREADDs) that selectively couple to either Gαq or Gαs. The central finding is that Gαq-stimulated gene expression, which includes a set of early response and inflammatory genes, is largely resistant to pharmacological BET inhibition with either the small molecule inhibitor JQ1 or the degrader dBET6. In contrast, the small subset of genes activated by Gαs signaling appeared to be sensitive to the BET degrader.

The study is well-conceived, and the data are generally of high quality. Also, the manuscript is clearly written and easy to read. The findings contribute to our understanding of the context-specific nature of BET protein function. However, there are several key areas where the study could be strengthened to make its conclusions more robust.

1. The most significant concern is the weakness of the Gαs-DREADD-induced transcriptional response. The RNA-seq analysis only identified four significantly upregulated genes upon Gαs activation.  This is a very small number upon which to build a major conclusion of the paper. The authors should conduct additional experiments to investigate if the experimental conditions were optimal for detecting a Gαs response. 

2. The study reports the unexpected finding that JQ1 treatment increases c-myc transcript levels, a phenomenon they also observed for a larger set of genes in their RNA-seq data, including HEXIM1. Although the proposed mechanism—release of P-TEFb from an inactive state upon Brd4 displacement from chromatin—seemed plausible and supported by literature, this complicates the interpretation of JQ1 as a simple "inhibitor" in this system. This paradoxical activation by JQ1 could be an unexpected factor in the experiments designed to test for BET "insensitivity." This reviewer would suggest authors to provide direct evidence that JQ1 displaces Brd4 from chromatin in their system. Maybe a ChIP-qPCR experiment?

3. HEK 293 cells were used throughout the study. However, its signaling network and transcriptional landscape may not be representative of all cell types, particularly the primary cardiomyocytes from their previous work. This point should be acknowledged somewhere in the manuscript.

4. In Figure 2A, the anti-BRD4 immunoblots are reported as normalized to the GAPDH loading control. However, in Figure S1A, β-tubulin was used as the loading control. Please clarify the rationale for changing the loading control and ensure consistency in figure labeling.

5. In Figure 2A, why was the Brd4 expression increased upon 2h dBET6 treatment? It is suggested to use longer time-course (for example, including results at 4h treatment) to verify the Brd4 expression decrease is stable.

6. In Figure 4D, the Venn diagrams are described in the text as “left” and “right,” but in the figure they are arranged vertically (“top” and “bottom”). This should be corrected for clarity and consistency.

7. In Figure 4D, the y-axis label is not readable due to low image resolution. Please provide a higher-resolution version of the figure so that axis labels are clear.

Reviewer 2 Report

Comments and Suggestions for Authors

Bromodomain and extraterminal domain (BET) proteins are dynamic regulators that are highly responsive to inputs from cellular signaling pathways. In previous work the authors showed that in rat cardiomyocytes, GPCR-induced hypertrophy response depended on the BET family protein Brd4 when signaling was coupled to of Gα isoforme Gαs , but not Gαq.

The manuscript, “Gαq-stimulated gene expression is insensitive to Bromo extra terminal domain inhibitors in HEK 293 cells” by Ashika Jain et al. devoted the investigation of study of differences in responsivity to G protein isoforms (Gαq and Gαs) in HEK 293 cells by expressing Gαs or Gαq-coupled Designer Receptors Exclusively Activated by Designer Drugs. Authors used BRET and RNA-seq data analysis for investigation of selectivity of G protein isoforms coupling and suggested that could be general signaling requirements to activate Brd4 across cell types (cardiomyocytes and HEK 293 cells particularly). It was shown that transcriptional induction by Gαq signaling as largely independent of BET coregulators. 

In my opinion, this is a good manuscript with interesting results. 

Minor revisions:

It recommend replacing in G protein isoforms Gas and Gaq “a” to “α”.

Line 381 “Thirty mg of supernatant was subjected to SDS-PAGE…” and line 395 "One mg of total RNA..."  "mg" should be replaced to "mkg".

Round 2

Reviewer 1 Report

Comments and Suggestions for Authors

All of my concerns have been addressed. I have no further concerns. Congratulations to the authors!